# Ultrasound Parameters Can Accurately Predict the Risk of Malignancy in Patients with “Indeterminate TIR3b” Cytology Nodules: A Prospective Study

**DOI:** 10.3390/ijms24098296

**Published:** 2023-05-05

**Authors:** Valentina Guarnotta, Roberta La Monica, Vincenza Rita Ingrao, Claudia Di Stefano, Riccardo Salzillo, Giuseppe Pizzolanti, Antonino Giulio Giannone, Piero Luigi Almasio, Pierina Richiusa, Carla Giordano

**Affiliations:** 1Endocrinology and Diabetology Section, Department of Health Promotion, Mother and Child Care, Internal Medicine and Medical Specialties “G. D’Alessandro”, PROMISE, University of Palermo, 90127 Palermo, Italy; 2Pathologic Anatomy Unit, Department of Health Promotion, Mother and Child Care, Internal Medicine and Medical Specialties, University of Palermo, 90127 Palermo, Italy; 3Gastroenterology and Hepatology Section, Department of Health Promotion, Mother and Child Care, Internal Medicine and Medical Specialties “G. D’Alessandro”, PROMISE, University of Palermo, 90127 Palermo, Italy

**Keywords:** follicular lesion, hypoechoic nodule, thyroid nodule, thyroid cancer, BRAF, ultrasound markers

## Abstract

The increase in the incidence of thyroid nodules with cytological findings of TIR3b requires the identification of predictive factors of malignancy. We prospectively evaluated 2160 patients from January 2018 to June 2022 and enrolled 103 patients with indeterminate cytology TIR3b nodules who underwent total (73 patients) and hemi-thyroidectomy (30 patients). Among them, 61 had a histological diagnosis of malignancy (30 classic papillary thyroid carcinoma, 19 had follicular papillary thyroid carcinoma variant, 3 had Hurtle cell carcinoma and 9 had follicular thyroid carcinoma), while 42 had a benign histology. Clinical, ultrasonographic and cytological characteristics were recorded. In addition, BRAF mutation was analysed. Patients with a histological diagnosis of malignancy had a higher frequency of nodule diameter ≤11 mm (*p* = 0.002), hypoechogenicity (*p* < 0.001), irregular borders (*p* < 0.001), peri- and intralesional vascular flows (*p* = 0.004) and microcalcifications (*p* = 0.001) compared to patients with benign histology. In contrast, patients with benign histology had more frequent nodules with a halo sign (*p* = 0.012) compared to patients with histological diagnosis of malignancy. No significant differences were found in BRAF mutation between the two groups. Our study suggests that the combination of ultrasonographic and cytological data could be more accurate and reliable than cytology alone in identifying those patients with TIR3b cytology and a histology of malignancy to be referred for thyroidectomy, thus reducing the number of patients undergoing thyroidectomy for benign thyroid disease.

## 1. Introduction

Thyroid nodules are a very common clinical problem. Epidemiological studies have shown that the prevalence of palpable thyroid nodules is approximately 5% in women and 1% in men [1,2], which increases to 19–68% with the use of high-resolution ultrasound (US) [3,4].

The clinical importance of thyroid nodules lies in the exclusion of a new diagnosis of thyroid cancer, notably in young patients, an event that occurs in 7–15% of cases depending on age, sex, history of radiation exposure, family history and other factors [5,6].

Indeed, in older patients or those with another, more life-threatening disease, the additional study of thyroid nodules is not needed.

Ultrasound is considered the method of excellence in the stratification of the risk of malignancy of the thyroid nodule and allows the operator to select the nodules to be subjected to fine needle aspiration, even though it has non-negligible limits such as low specificity and inter-operator variability [7,8].

Thyroid fine needle aspiration (FNA) cytological examination is the most accurate test for determining the thyroid nodule risk of malignancy [8].

The Italian classification of thyroid cytology was updated and published in 2014, making it comparable with the most widely used classifications, namely, the American one, known as the “Bethesda System”, and the English one of the Royal College of Pathologists of the United Kingdom (UKRCP). This classification maintains the five-category scheme associated with the suggestion of clinical behaviour in relation to the expected risk of malignancy. The TIR3 category (“indeterminate/follicular proliferation”) has been divided into two subclasses: TIR3a, or low-risk indeterminate lesion, which has the indication of repeating FNA since the expected risk of malignancy is <10%, and TIR3b, which has surgery as a priority option due to the expected risk of malignancy of 15 to 30%. This subclassification is intended to reduce the number of patients included in the TIR3 category undergoing surgery for benign disease. In the Bethesda and UKRCP classifications, these lesions are included in lower risk categories and need an FNA repetition at follow-up [9,10,11].

As highlighted in a recent meta-analysis of retrospective studies, rates of thyroid cancer in nodules with indeterminate low- and high-risk cytology often differ from those expected by the ICCRCT (Italian Consensus for the Classification and Reporting of Thyroid Cytology) classification, being, respectively, 17% and 47% against 10% of the expected risk of malignancy for TIR3a and 15–30% of TIR3b [12]. A recent meta-analysis evaluated the prevalence and risk of malignancy of the thyroid nodules with indeterminate FNA according to the Italian Classification System, estimating that the prevalence of cancer among TIR3b was 44.4% [13]. Nodules with at least one of the suspicious US features, such as marked hypoechogenicity, non-oval shape, irregular borders and microcalcifications, are considered by the ATA guidelines and the European classification of the Thyroid Imaging Reporting and Data System (EU-TIRADS) as a high-risk category of malignancy (26–87%) [7,14].

The aim of the current study was to analyse clinical and ultrasonographic variables, potentially predicting malignancy in patients with a cytological diagnosis of TIR3b in order to select patients who are candidates for surgery.

## 2. Results

A total of 61 out of 103 patients (59.2%) [17 males (27.9%) and 44 females (72.1%)] had a histological diagnosis of malignancy, while 42 patients (40.8%) [(10 males (23.8%) and 32 females (76.2%)] had a histological diagnosis of benignity (Table 1).

There were no statistically significant differences between the two groups with regard to gender, exposure to ionising radiation, smoking, a family history of thyroid carcinoma and benign nodular disease, autoimmune thyroiditis, pre-intervention therapy, BMI, waist circumference and calcitonin (Table 1 and Table 2).

Patients with a histological diagnosis of malignancy had a higher frequency of US characteristics, including nodular diameter ≤11 mm (*p* = 0.002), irregular borders (*p* < 0.001), nodule hypoechogenicity (*p* < 0.001), peri- and intralesional vascular flows (*p* = 0.004) and microcalcifications (*p* = 0.001), and a lower frequency of halo sign (*p* = 0.012) compared to patients with benign histological diagnosis (Table 1). In addition, patients with malignant histology had higher TSH values (*p* = 0.041) compared to patients with benign histology (Table 2).

Among patients with histological malignancy, 30 (49%) had a classic papillary thyroid carcinoma (PTC) variant, 19 (31.1%) had follicular PTC variant, 3 (4.91%) had Hurtle cell carcinoma and 9 (14.7%) had follicular thyroid carcinoma (FTC) (Figure 1A). Among patients with histological results of benignity, 33 (78.5%) had a follicular adenoma, 3 (7.14%) had Hurtle cell adenoma and 6 patients (14.3%) had non-invasive follicular neoplasm of the thyroid gland with papillary nuclear features (NIFTP) (Figure 1B).

Figure 2 and Figure 3 show a cytological specimen and histological examinations of benign and malignant tumours, respectively (Figure 2 and Figure 3).

### 2.1. Univariate Analysis

We calculated the nodule diameter and TSH cut-off values to distinguish patients with malignant diagnosis from benign histological diagnosis using receiver operating characteristic (ROC). We found a cut-off of nodule diameter ≤11 mm (sensitivity 39.3%, specificity 88.1%, area under curve 0.672) and a cut-off of TSH > 2.02 microU/mL (sensitivity 41%, specificity 81%, area under curve 0.693).

With univariate analysis, the nodule diameter ≤11 mm (OR 4.8, IC 1.65–13.9, *p* = 0.002), TSH > 2.02 microU/mL (OR 2.95, IC 1.17–7.43, *p* = 0.015) and US characteristics including irregular borders (OR 10.2, IC 3.87–27.1, *p* < 0.001), the hypoechogenicity of the nodule (OR 4.57, IC 1.95–10.64, *p* = 0.001), the presence of peri- and intralesional vascular flows (OR 2.57, IC 1.03–6.5, *p* = 0.034), microcalcifications (OR 4.69, IC 1.87–11.76, *p* = 0.001) and the absence of the halo sign (OR −1.66, IC 0.48–3.31, *p* = 0.002) were predictive risk factors of malignant histology (Table 3).

### 2.2. Multivariate Analysis

All significant univariate variables were included in the linear logistic regression model to rule out potential confounding factors. A multivariate analysis showed that a diameter ≤11 mm (adjusted OR 3.044, IC 1.10–10.8, *p* = 0.045; sensitivity 88.1%, specificity 39.3%, PPV 50% and NPV 82.8%), irregular borders (adjusted OR 4.98, IC 1.71–14.4, *p* = 0.003; sensitivity 83.3%, specificity 67.6%, PPV 63.6% and NPV 85.4%) and intranodular and perivascular flows (adjusted OR 3.46, IC 1.17–10.2, *p* = 0.025; sensitivity 81%, specificity 37.7%, PPV 47.2% and NPV 74.2%) were found to be predictors of malignant histology (Table 3). The Hosmer–Lemeshow test gave a *p*-value of 0.243, showing no evidence of poor fit, and correctly classified 75.7% of cases.

## 3. Discussion

Thyroid nodules with indeterminate (TIR3b) cytology represent a challenge for endocrinologists since the cytological characteristics alone are unable to predict the actual risk of malignancy of these lesions. Furthermore, although auxiliary molecular tests could contribute to more precise and personalised management, to date their use is limited due to their cost. In this study we used the SIAPEC/SIE 2014 classification for thyroid cytology, which identifies low-risk (TIR3a) and high-risk (TIR3b) indeterminate lesions [15]. These categories have different expected rates of malignancy (less than 10% for TIR3a and 15–30% for TIR3b), resulting in the general indication of follow-up for TIR3a and surgery for TIR3b nodules [9]. A recent study also evaluated the reproducibility of the SIAPEC/SIE classification, showing increased malignancy rates for both categories of TIR3 nodules [16].

In the current study, we aimed to evaluate whether the combination of US, molecular and clinical parameters may predict the risk of malignant histology in patients with indeterminate TIR3b nodules, in order to discriminate which patients should be candidates for surgery. We had a frequency of TIR3b of 4.7% with a malignancy rate of 59.2%, higher than expected by 15–30% [16]. However, many studies have reported a higher percentage of thyroid cancer in cytologically indeterminate TIR3b lesions, approximately 40% [15,17,18]. In addition, the higher malignancy rate of our study can be attributed to the higher incidence of thyroid cancer in Sicily [19,20].

We also showed that the presence of US characteristics, including nodule diameter ≤11 mm, irregular borders and peri- and intravascular flows, were strong predictors of malignancy in patients with TIR3b cytology. This finding is quite interesting because it is counter-intuitive. Indeed, in thyroid nodules, usually a large nodule size is associated with a higher risk of malignancy, but in TIR3b this was reversed. We may suggest that this was because there were mainly PTC among malignant tumours and mainly follicular tumours among benign ones. In the current study, all patients with TIR3b were suggested to undergo thyroid surgery. This indication is in line with that of the most recent guidelines for cytologically indeterminate nodules [16]. In addition, they expressed a preference towards surgery rather than active surveillance. A large number of the patients of the current study had undergone total thyroidectomy due to the presence of a multinodular goitre, even though hemi-thyroidectomy remains a good option for the treatment of patients with cytologically indeterminate nodules, even in cases of aggressive variants of papillary thyroid cancer [21].

In the current study, we observed a higher number of patients with classical PTC, rather than follicular variant of PTC, FTC and Hurtle cell carcinoma, in line with recent studies [12,22,23,24].

Our model was able to detect malignant nodules in three out of four patients. By contrast, the molecular testing for BRAF mutation did not show relevant significance, as expected, confirming the results of a recent study which showed that BRAF analysis had high specificity but poor sensitivity, including for cytological indeterminate thyroid nodules [25].

Our study identified a nodule diameter ≤11 mm as one of the predictive factors of malignancy. This finding is quite interesting because currently, in agreement with the most recent guidelines [16], sub-centimetric nodules should be subjected to FNA when suspicious ultrasound signs are present [8].

The most common US classification systems, the ATA guidelines and the TIRADS classification, have been widely demonstrated to be accurate for the identification of high-risk malignancy thyroid nodules [26,27]. Doppler criteria are currently controversial. Indeed, the intra-nodular vascularisation according to the European Guidelines is not included in the TIRADS score. The presence of an intra-nodular vascularisation in malignant nodules should only be considered as an additional potential risk; conversely, the absence of intra-nodular vascularisation should not be seen as a reassuring element [7].

With regard to the utility of US in the definition of malignant characteristics of thyroid nodules, Ianni et al. developed a score system to assess the risk of malignancy for thyroid nodules, the CUT score, which combined clinical and ultrasonographic parameters with the five categories of cytology of FNA [28]. Similarly, Rago et al. proposed a risk score based on the combination of three US characteristics including blurred borders, microcalcifications and hypoechogenicity and age < 40 years [29].

Several studies have tried to identify predictors of malignancy in patients with cytologically indeterminate lesions, showing discordant results. Some of them did not find any association between US parameters and malignant histology [30,31,32]. By contrast, a recent study combined the cytological subgroups with the ATA-based US classification of low, intermediate and high-risk, reaching 80% probability in the prediction of thyroid cancer [20]. Other studies have shown that the presence of irregular borders, microcalcifications, a shape “taller than wide” and hypoechogenicity were highly suggestive of malignancy in cytologically indeterminate TIR3 lesions [33,34]. Öcal et al. created a predictive model showing that the presence of microcalcifications and irregular borders were strongly suggestive of thyroid cancer in indeterminate thyroid nodules [35]. Interestingly, Amado et al. suggested that TSH values over 2.68 µU/mL were associated with a high risk of malignancy [36]. However, all the above-mentioned studies had a retrospective design.

In the current study, we found a significant difference in TSH preintervention values between patients with benign and malignant histology, even though in our opinion this difference did not have clinically significant relevance.

Another study evaluated how the rate of malignancy varied in each cytological category (from TIR2 to TIR5), taking into account gender, the echogenicity of the nodule and the presence or absence of intranodular chronic lymphatic thyroiditis (ICLT). In the TIR3b category, a rate over twice the expected 30% malignancy threshold was found in hypoechoic, ICLT-positive female nodules (80%), which contrasted sharply with the 38.5% rate of malignant tumours with the same characteristics but negative ICLT [37].

Only one recent prospective study evaluated the correlation between cytological, US and clinical features potentially associated with malignancy in patients with TIR3b cytology, demonstrating that parameters such as nodule size < 20 mm and age < 55 years were strong predictors of a malignant histology [38].

The relatively low number of patients included in the current study may be a limitation, and for this reason a comparative study of a larger sample is needed. Secondly, the use of the SIPAEC/SIE classification, which is frequently used in Italy, is not used very often in the rest of the world, hampering the generalisability of the results towards the rest of the world. The strength of our study is represented by the fact that it has a prospective design and by the analysis of the BRAF mutation in all patients. In addition, the cytology report was double-checked by two expert pathologists.

## 4. Materials and Methods

Our prospective study was conducted on a sample of 2160 patients referred to the Section of Endocrinology and Metabolic Diseases and Nutrition of the Policlinico Paolo Giaccone hospital, Palermo, between June 2018 and December 2022, who underwent FNA. The cytological examination showed a report of indeterminate lesion at a high risk of TIR3b malignancy in 103 out of 2160 patients (4.76%).

All patients with cytological diagnosis of TIR3b underwent total thyroidectomy or lobo-isthmectomy, as suggested by the current guidelines [16,39].

Patients were divided into two groups based on the histological diagnosis of malignancy and benignity. Patients with histological diagnosis of malignancy included cases of classic and follicular variants of PTC, Hurtle cell carcinoma and follicular carcinoma (Figure 1). The group of patients with benign histology included follicular adenoma, Hurtle cell adenoma and “Non-Invasive Follicular Neoplasia with Papillary Nuclear Features” (NIFTP) (Figure 2).

The clinical and US characteristics of the two patient groups were compared. A detailed medical history was collected in all patients, including smoking, history of previous irradiation to the neck and family history of thyroid cancer and/or benign thyroid disease, and presence of autoimmune thyroid disease. We also evaluated age, BMI, waist circumference, TSH and pre-FNA calcitonin values and US characteristics of the thyroid nodules (size, location, echogenicity, shape, vascular flow, presence of microcalcifications, characteristics of the borders and halo sign), according to the ATA guidelines [16].

The protocol of the study was approved by the local Ethics Committee of the Policlinico Paolo Giaccone, University of Palermo, and was carried out in accordance with the Helsinki declaration (1964 and later amendments) for human experimentation. All patients signed written informed consent, after a detailed explanation of the protocol of the study.

### 4.1. Fine Needle Aspiration Procedure

Ultrasound-guided thyroid FNA was performed using a 21–23 G gauge needle to aspirate nodules. All aspirates were smeared on at least 4–6 glass slides, and fixed and stained using May–Grunwald–Giemsa. Two expert cytopathologists working in the same institution analysed the FNA specimens and provided the cytological reports.

### 4.2. BRAF Mutation Analysis

After a FNAB procedure was performed, the needle was washed with 1 mL of sterile PBS and collected in a sterile tube. The tube was centrifuged at 800× *g* and the pellet extracted with the QIAamp DNA Micro Kit (Qiagen, Milan, Italy) following the manufacturer’s instructions, and DNA was eluted in 20 µL. Samples were submitted to real-time PCR, as previously described [40]. Briefly, the real-time PCR reaction was performed in a volume of 20 µL containing 10 µL of 2× QuantiNova Probe PCR master mix (Qiagen, Milan, Italy), 0.4 µM of each primer (BRAF-51F and BRAF-176R) and 0.2 µM of each FAM-labelled V600E probe and VIC-labelled wild-type probe and 3 µL of extracted DNA. Amplification was performed on a RotorGene Q (Qiagen, Milan, Italy) with the following conditions: 2 min at 95 °C for Taq activation, followed by 45 cycles of 15 s at 95 °C for denaturation and 30 s for annealing and extension. Fluorescence in the FAM and VIC channels were collected after the annealing–extension step. A wild-type sample and a V600E-carring sample were run along with unknow samples.

### 4.3. Statistical Analysis

SPSS version 17 and MedCalc version 11.3 were used for data analysis. Baseline characteristics were presented as mean ± SD for continuous variables; rates and proportions were calculated for categorical data.

The Shapiro–Wilk test was performed to evaluate the normality distribution of continuous variables. The differences between patients with histologies of malignancy and benignity were discovered using the Student’s *t*-test for continuous variables and with the chi-square test for categorical variables. Nodule diameter and TSH value were analysed by receiver operating characteristic (ROC) analysis to determine the cut-off to distinguish patients with malignant from benign histological diagnosis. The optimal cut-point value was defined as the value whose sensitivity and specificity were the closest to the value of the area under the ROC curve, and the absolute value of the difference between the sensitivity and specificity values was minimum. A univariate analysis was performed to evaluate the association between TIR3b and malignant histological diagnosis using the odds ratio (OR) and 95% confidence interval. Adjusted ORs were calculated by linear logistic regression and multivariate analysis to identify risk factors for malignancy independently associated with TIR3b. Only the factors that showed statistical significance at the univariate were included in the logistic regression analysis for multivariate analysis. A Hosmer–Lemeshow test was performed to evaluate the goodness of fit. A *p* value of <0.05 was considered statistically significant.

## 5. Conclusions

In conclusion, our study suggests that the combination of US and cytological data may be more accurate and reliable than cytology alone in identifying those patients with TIR3b and malignant thyroid disease to be referred for thyroidectomy. However, a larger multicentric prospective study is required to confirm our data.

## Figures and Tables

**Figure 1 ijms-24-08296-f001:**
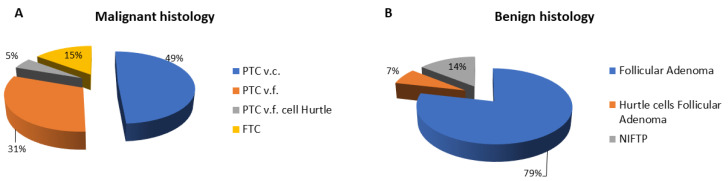
Distribution of malignant (**A**) and benign (**B**) histological diagnoses of cytologically indeterminate thyroid nodules after surgery. Abbreviations: PTC v.c., papillary thyroid cancer variant classical; PTC v.f., papillary thyroid cancer variant follicular; FTC, follicular thyroid cancer; NIFTP, non-invasive follicular thyroid neoplasm papillary-like nuclear features.

**Figure 2 ijms-24-08296-f002:**
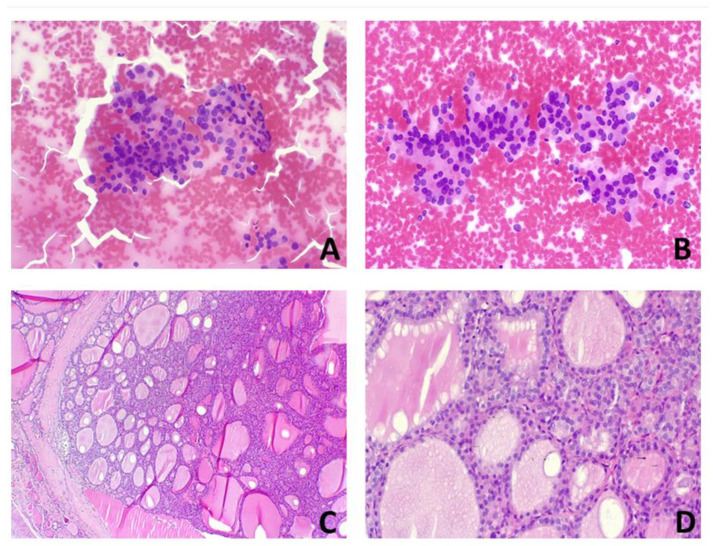
A case of TIR3b with benign histological examination: 70-year-old man. (**A**,**B**) Cytological examination showed moderate cellularity on a background with poor colloid; there are irregular aggregates of thyrocytes characterised by nuclear dysmetria and rare incisures. (**C**,**D**) Histological examination of total thyroidectomy showed follicular adenoma, circumscribed by a fibrous pseudocapsule with no signs of invasion.

**Figure 3 ijms-24-08296-f003:**
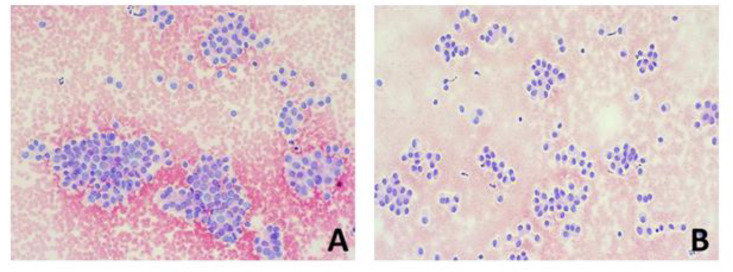
A case TIR3b with malignant histological examination: woman aged 41 years. (**A**,**B**) The cytological examination of the nodular lesion shows a blood background with poor colloid, including thyrocytes aggregated in irregular flaps and characterised by mild cytological atypia, with dysmetric nuclei with rare nuclear incisures. (**C**,**D**) Histological examination of thyroidectomy resected showed papillary carcinoma, follicular variety, bifocal. The neoplasm consists of cellular elements aggregated in follicular and, more rarely, papillary structures, characterised by crowded, overlapping, clarified nuclei and with incisions of the nuclear membrane. Immunohistochemistry showed positivity for HBME1 (**E**), CK19 (**F**) and Galectin3 (**G**).

**Table 1 ijms-24-08296-t001:** Ultrasound and clinical-anamnestic parameters in patients with histological outcomes of malignancy and benignity.

	Benign Histology (N = 42)	Malignant Histology (N = 61)	*p*
	**Subjects (%)**	**Subjects (%)**	
**Gender**			0.411
Male	10 (23.8%)	17 (27.9%)	
Female	32 (76.2%)	44 (72.1%)	
**Exposure to ionising radiation**			
	2 (4.8%)	3 (4.69%)	0.672
**Smoking**	14 (33.3%)	26 (42.6%)	0.229
**Familial history of thyroid cancer**			
	3 (7.1%)	7 (11.5%)	0.354
**Familial history of benign nodular pathology**			
	24 (57.1%)	29 (47.5%)	0.225
**Autoimmune thyroiditis**			
	10 (23.8%)	17 (27.9%)	0.411
**Ultrasonographic features**			
**Nodule diameter**			0.002
**≤11 mm**	5 (11.9%)	24 (39.3%)	
**>11 mm**	37 (88.1%)	37 (60.7%)	
**Irregular borders**	7 (16.6%)	41 (67.2%)	<0.001
**Echogenicity of nodule**			<0.001
Hypoechoic	13 (31%)	41 (67.2%)	
Isoechoic	12 (28.6%)	3 (4.9%)	
Iso-hypoechoic	14 (33.3%)	14 (23%)	
Complex	3 (7.1%)	1 (1.6%)	
Hyperechoic	0	2 (3.3%)	
**Vascular Flow**			0.004
Absent	18 (42.9%)	9 (14.8%)	
Perinodular	17 (40.5%)	30 (49.2%)	
Intra and perinodular	8 (19%)	23 (37.7%)	
**Microcalcifications**	8 (19%)	32 (52.5%)	0.001
**Taller than wide**	1 (2.4%)	2 (3.3%)	0.638
**Halo sign**	9 (21.4%)	3 (4.9%)	0.012
**Mutated BRAF V600E**	0	2 (3.2%)	
**Multinodular goitre**	28 (66.7%)	38 (62.3%)	0.405
**Pre-intervention therapy**			
Levothyroxine	5 (11.9%)	11 (18%)	0.289
Methimazole	1 (2.4%)	2 (3.3%)	0.638

**Table 2 ijms-24-08296-t002:** Clinical and hormonal parameters in patients with histological examination of malignancy and benignity.

	Benign Histology (N = 42)	Malignant Histology (N = 61)	*p*
	Mean ± SD	Mean ± SD	
**Age (years)**	56.1 ± 14.5	52.9 ± 14.1	0.267
**BMI (kg/m^2^)**	26.4 ± 4.16	27.2 ± 4.86	0.389
**Waist circumference (cm)**	92.1 ± 12.7	94.7 ± 14.2	0.341
**TSH pre-intervention (µU/mL)**	1.5 ± 0.81	1.87 ± 1.01	0.041
**Calcitonin pre-intervention (pmol/L)**	2.05 ± 1.38	2.78 ± 3.24	0.126

**Table 3 ijms-24-08296-t003:** Univariate and multivariate analysis of risk factors associated with malignant histology.

Variables	Benign Histology (N° = 42)	Malignant Histology (N° = 61)	Crude OR (95% CI)	Adjusted OR (95% CI)	*p*
**Diameter**					
>11 mm	5 (11.9%)	24 (39.3%)	1	1	
≤11 mm	37 (88.1%)	37 (60.7%)	4.8 (1.65–13.9)	3.044 (1.10–10.8)	0.045
**TSH**					
≤2.02	34 (81%)	36 (59%)	1		
>2.02	8 (19%)	25 (41%)	2.95 (1.17–7.43)		
**Irregular borders**					
No	35 (83.3%)	20 (32.8%)	1	1	
Yes	7 (16.7%)	41 (67.2%)	10.2 (3.87–27.1)	4.98 (1.71–14.4)	0.003
**Hypoechogenicity of the nodule**					
No	29 (69%)	20 (32.8%)	1		
Yes	13 (31%)	41 (67.2%)	4.57 (1.95–10.64)		
**Intranodular and perivascular flows**					
No	34 (81%)	38 (62.3%)	1	1	
Yes	8 (19%)	23 (37.7%)	2.57 (1.13–6.5)	3.46 (1.17–10.2)	0.025
**Microcalcifications**					
No	34 (81%)	29 (47.5%)	1		
Yes	8 (19%)	32 (52.5%)	4.69 (1.87–11.76)		
**Halo sign**					
Yes	9 (21.4%)	3 (4.9%)	1		
No	33 (78.6%)	58 (95.1%)	−1.66 (−3.31–0.48)		

## Data Availability

The data presented in this study are available on request from the corresponding author. The data are not publicly available due to patients’ privacy.

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
