# Peer review of "Ultrasound Parameters Can Accurately Predict the Risk of Malignancy in Patients with “Indeterminate TIR3b” Cytology Nodules: A Prospective Study"

_ijms, 2023, doi:10.3390/ijms24098296_

Round 1

Reviewer 1 Report

The review of  the paper “Ultrasound parameters can accurately predict the risk of malignancy in patients with "Indeterminate TIR3b" cytology nodules: a prospective study”.

The Authors concentrated on a group of 103 patients with thyroid nodules of indeterminate cytology (TIR3B). They analyzed the association between multiple features (ultrasound , clinical, hormonal, and BRAF V600E status), and malignancy of the nodule. They found multiple features that significantly differ between malignant and benign tumors. They also performed multivariate analysis in which logistic regression model was created. The model gives the accuracy of 75.7% when malignant vs benign classification was performed. The topic of discriminating benign and malignant thyroid tumors before surgery is important. The paper is clear, concise, and good written. I recommend the paper for publication in IJMS.

Major issues

1.       The Authors found that in TIR3b, malignant tumors are smaller than the benign ones. Can you discuss that counter-intuitive finding more? In thyroid nodules, usually the large nodule size is associated with higher risk of malignancy. But in TIR3b it is reversed. Is it because  there are mainly PTC among malignant tumors and mainly follicular tumors among benign ones?

2.       The Authors calculated that the accuracy of classification is 75.7% (line 150). Please, can you also calculate other performance measures of classification, like sensitivity, specificity, PPV (positive predictive value), NPV (negative predictive value) and area under ROC?

3.       The Authors determined the cut-off values of nodule diameter and TSH using ROC analysis. What was the criterion for determination of best cut-off?

4.       Did the Authors evaluate taller-than-wide nodule’s shape? If yes, please, include it in the paper.

Minor issues

5.       In the text, there are references to the Table 3 (for example in line 142), but there is no table 3 in the paper.

6.       There are some issues with abbreviations in the paper. For example WC abbreviation (table 2) is not explained, or PTC is explained in line 227 but used before.

7.       There is probably a misspelling in line 141: “OR -1.66, IC 0.48- -3.31, p=0.002” (to many minus signs).

8.       There is probably a misspelling in line 196: tIR3 (it should be TIR3).

Author Response

The Authors concentrated on a group of 103 patients with thyroid nodules of indeterminate cytology (TIR3B). They analyzed the association between multiple features (ultrasound , clinical, hormonal, and BRAF V600E status), and malignancy of the nodule. They found multiple features that significantly differ between malignant and benign tumors. They also performed multivariate analysis in which logistic regression model was created. The model gives the accuracy of 75.7% when malignant vs benign classification was performed. The topic of discriminating benign and malignant thyroid tumors before surgery is important. The paper is clear, concise, and good written. I recommend the paper for publication in IJMS.

Major issues

  1. The Authors found that in TIR3b, malignant tumors are smaller than the benign ones. Can you discuss that counter-intuitive finding more? In thyroid nodules, usually the large nodule size is associated with higher risk of malignancy. But in TIR3b it is reversed. Is it because there are mainly PTC among malignant tumors and mainly follicular tumors among benign ones?

Thanks for the comment. As you kindly pointed, it is a counter-intuitive finding. We added this comment in the text in the discussion section.

  1. The Authors calculated that the accuracy of classification is 75.7% (line 150). Please, can you also calculate other performance measures of classification, like sensitivity, specificity, PPV (positive predictive value), NPV (negative predictive value) and area under ROC?

Thanks for the comment. We added this information in the results section.

  1. The Authors determined the cut-off values of nodule diameter and TSH using ROC analysis. What was the criterion for determination of best cut-off?

Thanks for the comment. The optimal cut-point value was defined as the value whose sensitivity and specificity are the closest to the value of the area under the ROC curve and the absolute value of the difference between the sensitivity and specificity values is minimum. We added this information in the Statistical analysis subsection.

  1. Did the Authors evaluate taller-than-wide nodule’s shape? If yes, please, include it in the paper.

Thanks for the question. We added this information in the table, as you kindly suggested.

Minor issues

  1. In the text, there are references to the Table 3 (for example in line 142), but there is no table 3 in the paper.

Thanks for your right observation. We added the table 3 in the manuscript.

  1. There are some issues with abbreviations in the paper. For example WC abbreviation (table 2) is not explained, or PTC is explained in line 227 but used before.

Thanks for the comment. We defined the abbreviations in the text, as you kindly suggested.

  1. There is probably a misspelling in line 141: “OR -1.66, IC 0.48- -3.31, p=0.002” (to many minus signs).

Thanks for the comment. We corrected the value, as you kindly suggested.

  1. There is probably a misspelling in line 196: tIR3 (it should be TIR3).

Thanks for the comment. We corrected the misspelling, as you.

Reviewer 2 Report

The authors (prospectively) investigated whether combined ultrasound and clinical variables can discriminate between benign and malignant thyroid nodules which were TIR3b on cytology. Main outcome is that a such a combination could be of additional value in TIR3b lesions. After reading the manuscript several important issues remain:

1.      General comments

A)    The included patients are  those with TIR3b on cytology. As this Italian system is not used very often in the rest of the world, this hampers generalizability of the results towards the rest of the world. I suggest to add this to the discussion section.

B)    Almost 50% of the patients had classical PTC, which is a rather high percentage in TIR3b lesions which are usually FTC, FVPTC or Hurthle Cell Carcinomas (as also stated in Line 226-230). Can you explain this?

2.      Abstract

A)    Line 20 ‘thyroidectomy’; is this total or hemi?

B)    Line 21 ‘malignancy’; which type of thyroid cancer?

C)    Line 25 ‘By contrast’; one would think of changing into ‘In contrast’.

3.      Introduction

A)    Line 41 ‘need to exclude’; I wonder whether there is always this need, as one think of older patients or patients with another more life-threatening disease in which additional analysis is not needed.

B)    Line 77-78 ‘thus reducing invasiveness and cost and impact on quality of life, ensuring disease-free survival’; I feel that this is a very bold statement, and not something that is investigated upon in this research. I would leave this out. For example, there are numerous studies out there showing very little tumor increase in those on active surveillance in those with thyroid cancer. Besides, thyroid surgery could also lead to several complications (hoarseness, hypoparathyroidism) resulting into a decreased quality of life.

4.      Methods

A)    Line 271/272 ‘univariate analysis’; is this also logistic regression?

B)    Line 276 ‘the logistic regression analysis’; I would suggest to add ‘multivariate’ to this sentence.

5.      Results

A)    Line 98, those with malignant tumors have higher TSH levels than those with benign tumors. Were patients with low TSH excluded? Further, the difference was 0.37; is this also clinical significant?

6.      Discussion

A)    Line 167-168 ‘Our model was able……patients’. The problem with this statement is that I can’t find the model that is created, and secondly in this case there should be an external validation to really show the potential of this ‘model’.

B)    Line 168/169  ‘BRAF mutation did not show relevant significance’, which is a known phenomenon in patients with FTC, FVPTC or Hurthle Cell Carcinomas. Therefore, this is not very surprisingly.

C)    Line 172-175 Small nodule size leads to a higher risk of malignancy, especially in those ≤ 11mm. Could the authors explain why this is. Besides, why would you perform surgery in such small tumors, and not just do active surveillance? Next to this, how were the thyroid lesions initially discovered, as recent literature suggests that the risk of recurrence is significantly higher when thyroid cancer was diagnosed after a palpable mass was found (Lee et al. Cancers 2022).

D)    Line 213-214 ‘The strength……prospective design’. Why is the prospective design one of the strengths of the study. The major advantage compared to a retrospective study design in this case is probably that the chance of having missing data is lower in the prospective case, but are there any others in your opinion?

7.      Tables and Figures

A)    Table 2: please add explanation of the used abbreviations.

Author Response

The authors (prospectively) investigated whether combined ultrasound and clinical variables can discriminate between benign and malignant thyroid nodules which were TIR3b on cytology. Main outcome is that a such a combination could be of additional value in TIR3b lesions. After reading the manuscript several important issues remain:

  1. General comments
  2. A) The included patients are those with TIR3b on cytology. As this Italian system is not used very often in the rest of the world, this hampers generalizability of the results towards the rest of the world. I suggest to add this to the discussion section.

Thanks for this interesting comment. We added further information and the limits of the SIAPEC/SIE classification in the text.

  1. B) Almost 50% of the patients had classical PTC, which is a rather high percentage in TIR3b lesions which are usually FTC, FVPTC or Hurthle Cell Carcinomas (as also stated in Line 226-230). Can you explain this?

         Thanks for the question. In agreement with your comment, we think that the number of classical PTC in our cohort is very high. However, we reported other studies with increased percentage of PTC.

  1. Abstract
  2. A) Line 20 ‘thyroidectomy’; is this total or hemi?

Thanks for the comment. We added the information in the text.

  1. B) Line 21 ‘malignancy’; which type of thyroid cancer?

We defined the histotypes of thyroid tumors as you kindly suggested.

  1. C) Line 25 ‘By contrast’; one would think of changing into ‘In contrast’.

Thanks for the comment. We changed the sentence as you suggested.

  1. Introduction
  2. A) Line 41 ‘need to exclude’; I wonder whether there is always this need, as one think of older patients or patients with another more life-threatening disease in which additional analysis is not needed.

Thanks for the comment. We agree with you and clarified the sentence in the introduction section (lines 46-47)

  1. B) Line 77-78 ‘thus reducing

 invasiveness and cost and impact on quality of life, ensuring disease-free survival’; I feel that this is a very bold statement, and not something that is investigated upon in this research. I would leave this out. For example, there are numerous studies out there showing very little tumor increase in those on active surveillance in those with thyroid cancer. Besides, thyroid surgery could also lead to several complications (hoarseness, hypoparathyroidism) resulting into a decreased quality of life.

Thanks for the comment. We deleted this sentence accordingly to your suggestion.

  1. Methods
  2. A) Line 271/272 ‘univariate analysis’; is this also logistic regression?

Thanks for the question. It is not a logistic regression.

  1. B) Line 276 ‘the logistic regression analysis’; I would suggest to add ‘multivariate’ to this sentence.

Thanks for the comment. We clarified the text, accordingly to you suggestion (line 323)

  1. Results
  2. A) Line 98, those with malignant tumors have higher TSH levels than those with benign tumors. Were patients with low TSH excluded? Further, the difference was 0.37; is this also clinical significant?

Thanks for the comment. Patients with low TSH were not excluded. We add a comment on a not clinically significant relevance in the discussion section.

  1. Discussion
  2. A) Line 167-168 ‘Our model was able……patients’. The problem with this statement is that I can’t find the model that is created, and secondly in this case there should be an external validation to really show the potential of this ‘model’.

Thanks for the comment. As you kindly suggested we corrected the sentence, due to the absence of validation to define the predictivity of the model.

  1. B) Line 168/169 ‘BRAF mutation did not show relevant significance’, which is a known phenomenon in patients with FTC, FVPTC or Hurthle Cell Carcinomas. Therefore, this is not very surprisingly.

Thanks for the comment. We changed the sentence in the text as reported in line 211

  1. C) Line 172-175 Small nodule size leads to a higher risk of malignancy, especially in those ≤ 11mm. Could the authors explain why this is. Besides, why would you perform surgery in such small tumors, and not just do active surveillance? Next to this, how were the thyroid lesions initially discovered, as recent literature suggests that the risk of recurrence is significantly higher when thyroid cancer was diagnosed after a palpable mass was found (Lee et al. Cancers 2022).

Thanks for the comment. We add a comment in the text at lines 195-200.

  1. D) Line 213-214 ‘The strength……prospective design’. Why is the prospective design one of the strengths of the study. The major advantage compared to a retrospective study design in this case is probably that the chance of having missing data is lower in the prospective case, but are there any others in your opinion?

Thanks for the comment. In our opinion the main advantage of prospective study is the absence of missing data, as you actually reported.

  1. Tables and Figures
  2. A) Table 2: please add explanation of the used abbreviations.

Thanks for the comment. We explained the abbreviations as you kindly suggested.

Round 2

Reviewer 2 Report

The authors answered my questions and changed the manuscript. I have no further questions.